# Virus Infection and mRNA Nuclear Export

**DOI:** 10.3390/ijms241612593

**Published:** 2023-08-09

**Authors:** Jiayin Guo, Yaru Zhu, Xiaoya Ma, Guijun Shang, Bo Liu, Ke Zhang

**Affiliations:** 1University of Chinese Academy of Sciences, Beijing 100049, China; jyguo@ips.ac.cn (J.G.); yrzhu@ips.ac.cn (Y.Z.); xyma@ips.ac.cn (X.M.); 2Shanxi Provincial Key Laboratory of Protein Structure Determination, Shanxi Academy of Advanced Research and Innovation, Taiyuan 030012, China; gjshang@gmail.com; 3Key Laboratory of Molecular Virology and Immunology, Chinese Academy of Sciences, Shanghai 200031, China; 4Shanghai Huashen Institute of Microbes and Infections, Shanghai 200052, China

**Keywords:** virus, mRNA export, viral mRNA, host mRNA, NXF1, CRM1, Nup98, Rae1, NPC

## Abstract

Gene expression in eukaryotes begins with transcription in the nucleus, followed by the synthesis of messenger RNA (mRNA), which is then exported to the cytoplasm for its translation into proteins. Along with transcription and translation, mRNA export through the nuclear pore complex (NPC) is an essential regulatory step in eukaryotic gene expression. Multiple factors regulate mRNA export and hence gene expression. Interestingly, proteins from certain types of viruses interact with these factors in infected cells, and such an interaction interferes with the mRNA export of the host cell in favor of viral RNA export. Thus, these viruses hijack the host mRNA nuclear export mechanism, leading to a reduction in host gene expression and the downregulation of immune/antiviral responses. On the other hand, the viral mRNAs successfully evade the host surveillance system and are efficiently exported from the nucleus to the cytoplasm for translation, which enables the continuation of the virus life cycle. Here, we present this review to summarize the mechanisms by which viruses suppress host mRNA nuclear export during infection, as well as the key strategies that viruses use to facilitate their mRNA nuclear export. These studies have revealed new potential antivirals that may be used to inhibit viral mRNA transport and enhance host mRNA nuclear export, thereby promoting host gene expression and immune responses.

## 1. Introduction

The presence of a nucleus is the primary distinguishing factor between eukaryotic and prokaryotic organisms. The nucleus is a membrane-bound organelle found in eukaryotic cells that contains genetic material, including DNA and RNA, which directs the cellular activities of the organism. The nucleus is surrounded by the nuclear envelope, which is a double-layered membrane and acts as a physical barrier to protect the genetic material from damage and regulates the transport of molecules in and out of the nucleus. The nuclear pore complex (NPC) is a large protein complex that spans the double-layered membrane of the nuclear envelope in eukaryotic cells. The NPC acts as a gatekeeper, controlling the selective transport of molecules such as proteins, RNA, and other macromolecules between the nucleus and the cytoplasm [1,2,3].

To adapt to diverse stimuli and environmental cues, such as pathogen (viral, bacterial, or fungal) infections, cells must change their genetic expression and regulatory mechanisms. These changes include the activation or repression of specific genes through transcriptional regulation, as well as the processing, splicing, and degradation of mRNAs through post-transcriptional regulation. The resulting changes in gene expression can alter the production of proteins and other molecules that are critical for the cellular response to the stimulus or cue. By modulating gene expression, cells can adapt to changing conditions and mount an effective response to various challenges, including pathogen infections [4,5].

Regulations in innate immune responses to pathogens involve various molecular and cellular mechanisms that help a host defend against infection. These mechanisms include the recognition of pathogens by pattern recognition receptors (PRRs), the activation of immune signaling pathways, the transcriptional and post-transcriptional regulation of immune-related gene expression, and the regulation of nucleo-cytoplasmic trafficking. Nuclear export of mRNA, which encodes proteins for cellular defense, is an essential part of this regulation and allows the host to develop an effective response to pathogenic threats [6,7,8]. Viruses are considered to be one of the most significant types of pathogens due to their ability to cause a range of diseases in humans and other organisms. Compared with other pathogens, viruses are unique in their nature as they cannot reproduce or carry out metabolic processes without a host cell. They infect host cells and hijack their cellular machinery for their survival, with effects on their hosts ranging from trivial to deadly [9]. Viruses possess a variety of strategies to impair the host cell’s response to infection. As the nuclear export of host mRNAs is a critical step for the production of antiviral proteins, certain viruses have developed various strategies to effectively block host mRNAs nuclear export, including targeting transcription factors, mRNA processing pathway, and mRNA export factors to impair their functions in the nuclear export of mRNAs. With different genomic material, viruses can be divided into DNA virus and RNA virus. During infection, viruses replicate in the cells by hijacking cellular mechanisms, such as transcription, mRNA processing, mRNA transportation, and translation [5,9,10].

The cellular nuclear export machinery is used by most DNA viruses and a few RNA viruses whose replication occurs in the nucleus [11]. Studies of the mechanisms by which viruses export viral RNA have shown that two important nuclear export factors for host RNA, named CRM1 (chromosome region maintenance 1, also known as exportin 1 or XPO1) and NXF1 (nuclear export factor 1, also known as tip-associated protein or TAP) [12,13,14], are utilized by several types of viruses to promote the export of viral mRNA while also inhibiting the transport of host mRNA to prevent an appropriate host immune response [15,16].

The mRNA precursors (pre-mRNAs) must be properly processed into mature mRNAs by capping, splicing, and polyadenylation before being exported from the nucleus [17,18]. Following these processing, mRNA export factors start loading on the mRNA and draw the export receptor. The transcriptional export (TREX) complex coordinates the nuclear export of the mRNA via transcription and processing [19,20,21]. This complex consists of a multisubunit THO complex [20,22,23,24] that plays a role in transcription elongation, the export factor UAP56 (also known as HEL), and Aly/REF (also known as THOC4). UAP56 binds to Aly/REF and facilitates the binding of Aly/REF to mRNAs [25,26,27]. Aly/REF interacts with the main mRNA export receptor nuclear RNA export factor 1 (NXF1, also known as TAP)–Ntf2-like export factor 1 (NXT1, also known as TAP-p15), a heterodimer that mediates the nuclear export of mRNA by binding to the phenylalanine-glycine (FG) repeat domain of Nups [14,28,29,30]. Nups containing FG, such as Nup98, are translocated via NPC-mediated mRNP (messenger ribonucleoprotein) complexes [30,31]. Rae1 (also known as Mrnp41) is another mRNA export factor involved in this process. It is a nucleo-cytoplasmic shuttle protein that interacts with mRNA, NXF1, and Nup98 [32,33,34,35,36,37,38,39,40]. Rae1 can recruit NXF1 to Nup98 [34,41,42,43]. CRM1 is another important receptor which is known to mediate the nuclear export certain types of RNA, including miRNA, and mRNA subgroups [41,42,43]. RNA nuclear export via CRM1 requires an adaptor protein that binds to cargo RNA, which interacts with CRM1 via nuclear export signal (NES) [44].

Although many host cell processes are involved in viral replication and assembly, we discuss below the mechanisms by which viruses hijack or usurp the host’s mRNA export pathway to inhibit the host’s mRNA nuclear export, thereby reducing the expression of the host’s immune genes and causing a successful and sustained infection [45,46].

## 2. Cellular mRNA Nuclear Export

mRNA nuclear export, a key step before functional protein production, is a crucial process that plays an important role in gene expression regulation. In eukaryotic cells, transcription occurs in the nucleus, and the newly synthesized mRNA molecules must be exported to the cytoplasm for translation to occur. The export of mRNA from the nucleus to the cytoplasm involves a complex series of interactions between various transcription factors, such as activators/repressors and pre-mRNA processing factors, with TREX complex, transcription and export 2 (TREX-2) complex, and the NPC being the crucial elements necessary for mRNA transport [31,47,48].

The mechanism of mRNA nuclear export is facilitated by the NPC, which is a large protein complex that spans the double-layered membrane of the nuclear envelope in eukaryotic cells. The NPC acts as a gatekeeper, controlling the selective transport of molecules such as proteins, RNA, and other macromolecules between the nucleus and the cytoplasm. The NPC is a cylindrical channel that spans the nuclear envelope and is made up of numerous copies of around 30 different Nups, which are placed in an octagonal symmetry. The central channel of the NPC is lined with phenylalanine-glycine (FG) repeats, which form a selective barrier that allows only certain molecules to pass through. The NPC is critical for many cellular processes, including gene expression, RNA processing, and protein transport. The NPC also serves as a quality control mechanism for the exported mRNA molecules. The export process is mediated by a group of proteins, including exportins and TREX complex [49,50,51,52].

There are two main pathways for mRNA nuclear export: NXF1–NXT1-dependent and CRM1-dependent pathways. After capping, splicing, and polyadenylation, the pre-mRNA becomes mature mRNA [17,18]. Then, mRNA export factors begin to load on the mRNAs and recruit the TREX complex. In eukaryotic cells, the mRNA nuclear export process is significantly influenced by the TREX complex. The THO complex is responsible for the recognition and binding of mRNA molecules after they are synthesized [23]. In the ATP-bound condition, the UAP56 forms a complex with Aly/REF, leading to an increased affinity of Aly/REF for mRNA [25]. Subsequently, Aly/REF recruitsNXF1 -NXT1 to the mRNA [53]. The NXF1–NXT1 heterodimer is the major mRNA nuclear export receptor for the bulk of cellular poly(A)-RNA. NXF1 is the primary component of this complex, while NXT1 acts as an adaptor protein that mediates the interaction between NXF1 and mRNA. By binding to the phenylalanine-glycine (FG) repeat domains found on NPC proteins (nucleoporins or Nups), such as Nup62 and Nup98, the NXF1–NXT1 heterodimer facilitates the transport of mRNA through the NPC and into the cytoplasm for translation [29,30,31]. Through the interaction with GANP, NXF1 can associate with the TREX-2 complex, which includes GANP, PCID2, DSS1, ENY2, and CETN3. With a stable association with the nuclear pore basket, the TREX-2 complex has been found to play a role in the nuclear export of a subset of cellular mRNAs. GANP could interact with nucleoporin TPR on the nucleo-cytoplasmic side of NPC. Messenger ribonucleoprotein (mRNP) translocation via the NPC is mediated by the binding of NXF1-NXT1 with GANP during mRNA nuclear export [54,55,56].

The CRM1-dependent mRNA nuclear export pathway is another mechanism by which certain mRNAs are exported from the nucleus to the cytoplasm. This pathway is dependent on the activity of CRM1, a member of the karyopherin beta protein family that functions as an export receptor for certain NESs found on cargo proteins, including certain mRNAs, tRNAs, and viral RNAs. In this pathway, CRM1 does not directly bind to mRNA, but requires the assistance of RNA export adaptor proteins that can directly bind to the RNA molecules. These adaptor proteins contain NESs that are recognized by CRM1, which then interacts with RanGTP to facilitate the export of the RNA through the NPC [44,57].

In addition, for the host RNA export, RNA length plays an important role in determining which pathway the RNA takes. McCloskey et al. showed that the heterologous nuclear ribonucleoprotein (hnRNP) C tetramer has the function of measuring RNA length for the classification export of RNA polymerase II transcripts [58]. Dantsuji et al. demonstrated that hnRNP C tetramers play a key role in the identification and measurement of RNA length. The heterotetramer of the heterogeneous nuclear ribonucleoprotein (hnRNP) C1/C2 acts as a molecular ruler to measure the length of the transcripts by selectively binding to the unstructured RNA regions longer than 200 to 300 nucleotides. Thus, the tetramer sorts the transcripts into two RNA categories: mRNA or uridine-rich small nuclear RNA (U snRNA). Further mechanistic studies proved that the hnRNP-C tetramer binds the cap-binding complex (CBC) on mRNA and blocks PHAX (an adaptor protein for splicosomal U snRNA output) recruitment for the classification of RNA polymerase II transcripts, so as to direct these RNAs to the mRNA nuclear export pathway [59]. This discovery of the hnRNP C tetramer for length-specific RNA classification contributes to a deeper understanding of RNA biogenesis and provides potential strategies for future disease treatments.

In conclusion, mRNA nuclear export is a complex and highly regulated process that plays a vital role in gene expression regulation. Understanding the underlying mechanisms of mRNA nuclear export provides important implications for our understanding of cellular function and disease pathology.

## 3. Viral Infection Inhibits the Nuclear Export of Host mRNA

Viruses utilize the host mRNA nuclear export pathway and prevent the export of host mRNA, which is crucial for the translation of mRNA encoding cellular defense proteins required to generate a proper immune response against the invading pathogen. Thus, the mRNA export pathway is an attractive target for viruses to block the expression of host antiviral genes (Figure 1).

In order to block the expression of host genes, viral proteins can utilize host mRNA nuclear export mechanisms to interfere with the host mRNA nuclear export. Multiple viral proteins can be involved in this process. For instance, non-structural protein 1 (NS1) of influenza A virus (IAV) prevents the interaction between NXF1–NXT1 and the nucleoporins, thereby blocking the export of the host’s mRNA [60]. IAV NS1 also interacts with the mRNA nuclear export factor Rae1 and E1B-AP5 to inhibit host mRNA export [60]. Similarly, the M (matrix) protein of vesicular stomatitis virus (VSV) interacts with Rae1 and NPC (via NUP98) to form M-Rae1-NUP98, blocking host mRNA nuclear export [36,61]. The M proteins of other vesicle viruses, such as the Chandipura virus (CV) and spring viraemia of carp virus (SVCV), also block the export of host mRNA by interacting with Rae1 and NUP98 [62]. In addition, the adenovirus (AdV) protein E1B-55K inhibits the nuclear export of host mRNA by interacting with E1B-AP5. Moreover, E1B-55K is assisted by its E4orf6 protein, which can form [63] part of the E3 ubiquitin ligase with extender proteins B and C, cullin 5, RBX1, and other proteins of the host cell [63]. This ubiquitin ligase targets cellular p53 [64], MRN (MRE11–RAD50–NBS1 complex, involved in DNA double-strand break repair) [65], DNA ligase IV [66], and integrin α3 [67] to alter the function of host cells. Thus, viral proteins impair the export of host nuclear mRNAs through a variety of pathways by interacting with specific mRNA export factors.

In the downregulation of host mRNA nuclear export by viral proteases, viral proteins can also manipulate the host’s nucleo-cytoplasmic transport by degrading nucleoporins. For instance, the 2A protease (2APro) of picornaviruses, such as poliovirus (PV) and human rhinovirus (HRV), causes protein degradation in nucleoporins, including NUP62, NUP98, and NUP153, leading to changes in the NPC structure and inhibiting host mRNA nuclear export [68,69,70,71,72,73,74]. Similarly, HRV’s 3C protease (3CPro) is involved in the proteolytic degradation of NUP153, NUP214, and NUP358, thereby affecting NPC and mRNA export [75]. Unlike these viruses, picornaviruses, such as Theiler’s murine encephalomyelitis virus (TMEV) and encephalomyocarditis virus (EMCV), lack proteolytic activity in 2APro, but they still impact RanGTP’s involvement in CRM1-mediated mRNA export by the viral leader (L) proteins [11,76,77,78]. The interaction between L proteins and RanGTP also affects protein transport, as RanGTP and CRM1 are involved in protein transport.

In addition to targeting host mRNA nuclear export factors and nucleoporins, viruses downregulate host gene expression by destabilizing host mRNA. For example, viruses such as IAV and IBV (influenza B virus) target the 5’ cap of the host’s mRNA for its degradation [79]. The influenza virus’s RNA polymerase (vRdRp: viral RNA-dependent RNA polymerase) has three subunits, PA, PB1, and PB2, and is responsible for this process. PB2’s mRNA cap-binding domain binds to the host mRNA cap, and then PA’s endonuclease cleaves the host RNA at the 10th–13th nucleotide downstream of the cap. Viruses can also stabilize host mRNA to alter its gene expression. For example, Zika virus, an insect-borne RNA-containing flavivirus, produces long non-coding RNA (lncRNA) called sfRNAs, which are 300–400 nucleotides long. The repression of XRN1 (a 5’ to 3’ exoribonuclease) by sfRNA located in the 3‘ UTR leads to a reversible inhibition of enzyme activity, likely due to the slow release of the inhibited enzyme from the RNA substrate. The Zika virus sfRNA interacts with a common set of 21 RNA-binding proteins (such as DEAD-box helicase 6 (DDX6), mRNA uncapped enhancer 3 (EDC3), etc.) that help regulate cellular post-transcriptional processes, including splicing, RNA stability, and translation [80].

### 3.1. RNA Viruses Inhibit Host mRNA Nuclear Export

The influenza virus is an enveloped negative-strand RNA virus. The viral ribonucleoproteins (vRNPs), assembly by RNA-dependent RNA polymerase (RdRP) and nuclnucleoprotein (NP), is responsible for viral RNA transcription. In contrast to most negative-strand RNA viruses, influenza viruses transcribe their mRNA in the nucleus. Transcription is initiated by the RdRP excising 12–15-nucleotide RNAs from the mRNA of the infected cell in a process called “cap grabbing” [81] and polyadenylation by repeated replication occurs around the 5′ end of the vRNA template’s poly(U) stretch [82]. Two viral transcripts, M and NS, undergo alternative splicing. The nuclear export of various types of mRNAs, including spliced mRNAs (M2 and NS2), intron-containing mRNAs (M1 and NS1), and intron-free mRNAs (HA, NA, NP, PB1, PB2, and PA), is essential for viral replication. Similar to most viral infections, the proliferation of IAV within vertebrate cells gets detected by the innate immune system. After recognition, the innate immune system initiates signal transduction pathways that result in the synthesis of type I interferons (IFNs), a group of antiviral cytokines that stimulate the expression of mRNAs encoding antiviral factors [83], including nucleoporins [68,84]. In order to suppress host anti-viral response, IAVs have developed several strategies, primarily utilizing non-structural protein 1 (NS1), which stays mainly in the nucleus during early stage of infection, to inhibit mRNA processing and export [5]. There are close linkages between the mRNA processing and nuclear export since some proteins interacting with mRNA are involved in both processes, while others are replaced by particular factors [85]. By binding to the spliceosome component or interacting with the splicing factor, such as NS1 binding protein (NS1-BP), NS1 inhibits pre-mRNA splicing [86,87]. NS1 further disrupts the splitting and polyadenylation specific factor (CPSF30) and poly(A) binding protein II (PABII) of the host’s mRNA processing-bound 30 kDa subunit, participating in the binding polyadenylation signal and the extension of the poly(A) chain mRNA, respectively [88,89]. The interaction of NS1 with these proteins inhibits the 3’-terminal processing of host mRNA and helps to inhibit the expression of host genes. However, because viral polymerase complexes synthesize the poly(A) tail on viral mRNAs, NS1-mediated interruption of mRNA processing has no impact on the generation of viral transcripts [82,90]. In addition, by interacting with host mRNA-binding proteins, NS1-BP, and hnRNP K, NS1 facilitates the splicing of viral M1 mRNA fragments [91]. This enables the efficient processing of viral mRNA transcripts before nuclear export occurs. In addition to disrupting the host mRNA processing, IAV further downregulates the expression of host antiviral genes through NS1 interacting with the host mRNA nuclear export mechanism. NS1 directly interacts with the mRNA export factor NXF1–NXT1 (Figure 2A), which forms complexes with Rae1 and E1B-AP5, result in the retention of poly(A) RNA in the nucleus [60,92]. The NS1–NXF1–NXT1 complex is solved by crystallization. In the protein complex, two NS1 molecules bind to two NXF1–NXT1 heterodimers. The two NS1 molecules dimerize through their RNA binding domains. One effector domain binds to the LRR domain of NXF1. The phenylalanine 103 of the other effector domain binds to the NTF2L domain of the neighboring NXF1, which is also the nucleoporin binding domain, leading to the inhibition of NXF1 docking to NPC. Using the reverse genetic method, a mutant influenza virus carrying the NS1 mutations, in which key residues involved in the interaction with NXF1 were mutated, was generated. The NS1 mutant virus, which lost the interaction with NXF1, did not inhibit the host mRNA nuclear export, resulting in the release of an antiviral response. Therefore, the replication of the mutant virus was attenuated [16]. Taken together, these investigations propose that the IAV NS1 employs diverse strategies to impede the intricate and precisely controlled procedures of mRNA processing and export. Notably, NS1-mediated inhibition of mRNA nuclear export results in the retention of mRNAs encoding antiviral genes in the nucleus [93], indicating that the virus potentially disrupts these pathways as a strategy to enhance viral replication and evade host immune responses.

Belonging to the *Rhabdoviridae* family, Vesicular Stomatitis Virus (VSV) has a single-stranded RNA genome. VSV is known for causing vesicular stomatitis, a viral disease that affects cattle, horses, and pigs, as well as humans. The virus is transmitted through direct contact with infected animals, their saliva, or contaminated objects. Five genes are encoded by the VSV RNA genome, and they are organized in the following sequence: 3’-N-P-M-G-L-5’. The M gene encodes the matrix protein (M), which is necessary for virion assembly and budding. During VSV infection, the cellular mRNA nuclear export can be blocked by the interaction between M and RNA binding protein Rae1, a component of the mRNA export machinery, which forms a complex with nucleoporin Nup98 (Figure 2B) [36,61,94,95]. The interaction between VSV M protein and Rae1 leads to the sequestration of Rae1 in the nucleus, preventing its normal function in mRNA nuclear export. The participation of the VSV M protein in the inhibition of the host mRNA nuclear export has been detected by multiple methods, including poly(A) RNA-FISH, qPCR of nucleo-cytoplasmic fractionation samples, and nuclear export assays. The residue methionine 51 plays a key role in the interaction between M and Rae1. The M51R mutation of M loses its function in the mRNA nuclear export blockage [96]. The M–Rae1–Nup98 complex leading to a blockage of bulk mRNA nuclear export may be due to two reasons: the nuclear export of a subset of mRNAs that encode gene expression regulators that are blocked, or the Rae1-Nup98 complex involved in regulating the NXF1-mediated mRNA nuclear export. As a result, the expression of antiviral factors is downregulated [93]. The inhibition of the host’s mRNA export by VSV enables the virus to transport its limited number of mRNAs into the cytoplasm, preventing competition for the translation machinery with the host mRNAs. Overall, hindering cellular mRNA trafficking by M is one of the most important strategies to enable VSV to regulate host gene expression and promote its own replication [97].

Global public health has been significantly affected by the pandemic caused by severe acute respiratory syndrome coronavirus 2 (SARS-CoV-2) [98]. The SARS-CoV-2 genome is approximately 30 kb in length and consists of positive-sense single-strand RNA. It contains fourteen open reading frames, which encode a large polyprotein, four structural proteins (spike (S), membrane (M), envelope (E), and nucleocapsid (N)), and nine accessory proteins (ORF3a, ORF3b, ORF6, ORF7a, ORF7b, ORF8, ORF9b, ORF9c, and ORF10) [99]. SARS-CoV-2 hijacks the host cellular machinery to promote genome replication, which occurs in the cytoplasm. To favor replication, SARS-CoV-2 adopts multiple strategies to suppress the host immune and antiviral responses, including inhibiting host mRNA nuclear export. One of the identified mechanisms involves SARS-CoV-2 Nsp1. Nsp1 was shown to interact with the NXF1–NXT1 heterodimer, leading to the retention of cellular mRNAs in the nucleus. An immunoprecipitation assay with a nuclear fraction showed that Nsp1 displaces NXF1 from the NPC. mRNA inhibition by SARS-CoV-2 was reverted by the overexpression of NXF1 in vero-E6 cells, resulting in the reduction of SARS-CoV-2 replication [99]. SARS-CoV-2 may also target the Nup98–Rae1 complex by Orf6 (Figure 2C) to subvert nucleo-cytoplasmic trafficking, including protein import and nuclear export of mRNA transcripts, such as IFN-upregulated genes. The interaction between Orf6 and Nup98-Rae1 is greatly decreased by the M58R mutation [46,100,101]. Several other SARS-CoV-2 proteins may also interact with mRNA nuclear export factors or nucleoporins, indicating their unknown roles in regulating cellular mRNA nuclear export [102]. The underlying mechanisms that SARS-CoV-2 may use to inhibit host mRNA nuclear export need to be further studied.

### 3.2. DNA Viruses Inhibit Host mRNA Nuclear Export

AdVs, a non-envelope DNA virus, contains around 36 kbp long linear double-stranded DNA genome. They got their name because of the first discovery in adenoid cell cultures. AdVs are widely distributed, and infections in the immunocompetent host are typically mild or asymptomatic and resolve independently. Nonetheless, Advs infection can result in a more severe illness in people with compromised immune systems [103]. The genome of human AdVs is quite dense, with more than 40 genes. The total length of AdV genomes can vary among different serotypes, ranging from approximately 26,000 to 45,000 base pairs [104]. To ensure efficient viral replication, these genes are expressed in an orderly manner that allows the host cell to replicate to the virus without alerting the immune system or inducing cell damage that leads to apoptosis [105]. Early gene 1A (E1A) was the first gene to be expressed, and it promotes the production of other early proteins from E2, E3, and E4 transcription units [106]. Subsequently, the expression profile of Adv genes gradually changes to a major late promoter (MLP) upon infection, leading to the promotion of the expression of late gene products, which is responsible for the assembly of the new viral particle [106]. In general, a variety of mechanisms, including alternative splicing, which leaves specific transcripts partially or entirely unspliced, highly regulate the transcription program of AdV [107]. Additionally, specific viral transcripts are exported to the cytoplasm for expression at the appropriate time [106]. To date, studies on the export mechanism of AdV mRNA have focused on late messengers, and the picture appears to be quite complex. In 1986, the possible function of viral E1B-55k protein on AdV late transcript export was first observed [108]. However, it is believed to play a relatively indirect and upstream role in export, occurring before the initiation of mRNA nuclear export [109]. Instead, E1B-55K is the primary coordinator leading to the viral protein E4orf3, the promyelocytic leukemia (PML) nucleosome, viral transcription, and replicating occurrence [110,111]. During the later stages of infection, a different viral protein known as E4orf6, competes with E4orf3 to trigger the reorientation of E1B-55K towards the nuclear blob, which serves as the storage site for splicing factors, export capacity, and both viral and cellular transcripts [111]. These series of modifications induced by viruses in nuclear structure hinder the maturation and export of cellular mRNAs, while simultaneously facilitating the export of viral transcripts that include introns. [112]. In addition, to efficient nuclear export of viral late mRNA, the ubiquitin-3-ligase complexes formed by E1B-55k, E4orf6, and some cellular components are required [113]. The relevant mechanisms depend on the cellular components necessary to inhibit the degradation of intron-containing transcripts [114]. However, the understanding of the numerous other functions of E1B-55k in regulating AdV gene expression still need further investigation [115].

Kaposi’s sarcoma-associated herpesvirus (KSHV), also known as human herpesvirus 8 (HHV-8), is a nuclear DNA virus belonging to *Herpeseviridae*. KSHV is a complex virus with a double-stranded DNA genome that encodes for more than 80 genes. The virus has two distinct phases in its life cycle: a latent phase and a lytic phase. During the latent phase, the virus resides within the host cell without causing any harm, while during the lytic phase, the virus replicates and produces infectious particles. KSHV encodes several viral proteins that can modulate the host cell’s immune response [106]. The ORF10 gene, which can produce ORF10 protein, is located in the viral genome downstream of the viral interleukin-6 (vIL-6) gene and is expressed during the late phase of the viral life cycle, when viral particles are being assembled. Several studies with RNA-FISH and immunoprecipitation methods have suggested that ORF10 may play a role in immune evasion by the virus, including binding to the Rae1–Nup98 complex to selectively inhibit cellular mRNA nuclear export (Figure 2D). By solving the crystal structure of the ORF10–Rae1–Nup98 complex, Feng et al. showed that L60 and M413, two highly conserved residues, are critical for both the assembly of the Orf10-Rae1-Nup98 complex and the suppression of mRNA nuclear export. In vitro binding assays showed that although ORF10 occupies the RNA binding surface of Rae1–Nup98, the ORF10–Rae1–Nup98 complex still binds RNA through ORF10. When mutations are introduced on ORF10’s RNA-binding surface, the protein’s ability to inhibit mRNA export is disrupted. The ability of KSHV to selectively inhibit RNA export can limit host gene expression and optimize the facilitation of replication [116,117].

In sum, during virus infection, a significant strategy utilized by viruses is to trap host mRNA in the nucleus. This results in a downregulation of the host’s gene expression and the inhibition of antiviral responses, which allows the virus to evade host immune responses and establish a successful infection. Various viruses use distinct mechanisms to inhibit host mRNA nuclear export (as summarized in Table 1). Gaining a better understanding of the molecular mechanisms involved in this process could provide valuable insights for the development of novel antiviral therapies.

## 4. The Nuclear Export of Viral mRNA

The nuclear export of viral mRNA refers to the process by which viral mRNA is released from the nucleus. This process is the first step that viruses take to infect cells, and it can be achieved through the action of viral proteins. The viral protein will bind to the viral mRNA and release it from the nucleus so that the viral mRNA can enter the cytoplasm, where it can be transcribed and translated to produce new viral proteins.

Viruses usurp or hijack host mRNA nuclear export mechanisms to transport their own RNA for viral replication and assembly and block host mRNA export, thereby reducing the expression of host genes, including genes involved in immune responses (i.e., interferon (IFN), inducer genes, and NF-kb regulatory genes [123,124,125]) for successful viral infection and reproduction. By promoting its mRNA export, a virus can also deceive the host immune system by expressing genes that encode their own cytokines, molecules that neutralize host cytokines and homologues of TNF (tumor necrosis factor) or MHC (major histocompatibility complex) classes, acting as decoys for NK (natural killer) cells [123,124,125,126,127]. In addition, virus-encoded lncRNA (long non-coding RNA) has also been implicated in antiviral responses against IFN [128,129,130,131,132,133]. For example, the PANRNA (polyadenylated nuclear RNA) of KSHV (Kaposi sarcoma-associated herpes virus) reduces IFNα and IFNγ expression [128,129]. sfRNAs (subgenomic flavivirus RNA) of several viruses, including JEV (Japanese encephalitis virus), Dengue virus, and WNV (West Nile virus), inhibit IFNβ expression and IFN signaling, thereby impairing the host’s antiviral response [128,129,130,131,132,133]. In addition, by enhancing its RNA export and producing molecules that inhibit caspase (a major player in apoptosis), viruses can also block host cell apoptosis strategies to get rid of virus-infected cells [96]. Thus, viruses hijack the host’s mRNA export mechanism to help induce a successful infection and subsequent reproduction [123]. This hijacking is mediated through the interaction of viral proteins with host mRNA nuclear export factors.

### 4.1. The Virus Targets the Host mRNA Export Receptor CRM1

The nuclear export receptor CRM1 recognizes cargo proteins containing a leucine-rich NES. During virus infection, some viruses hijack the CRM1-dependent mRNA export pathway to export their own viral mRNA molecules out of the nucleus and into the cytoplasm, where they can be translated into viral proteins. Viruses can achieve this process by encoding NES-containing proteins, which are recognized by CRM1. The CRM1-mediated nuclear export pathway is a critical step in the replication cycle of many viruses, including human immunodeficiency virus 1 (HIV-1), prototype foamy virus (HFV), human T-cell leukemia virus type 1 (HTLV-1), and human papillomavirus (HPV) (Figure 3A). Targeting this pathway with specific inhibitors has been shown to inhibit viral replication, and could be a potential therapeutic strategy for viral infections.

HIV-1 and other lentiviruses use the host mRNA export receptor CRM1 to export their RNA into the cytoplasm. The major HIV-1-encoding proteins are Gag, Pol, Env, Tat, Rev, and Nef [134]. The genome replication of HIV-1 occurs in the nucleus. Thus, the HIV-1 mRNA should be exported to the cytoplasm for translation. The Rev protein contains a leucine-rich NES domain, which is recognized by CRM1. In addition, the Rev protein interacts with the RRE (Rev response element, a highly structured RNA), which is often unspliced. The Rev also acts as an arginine-rich NLS (nuclear location signal) [10,135,136,137]. This viral RNA Rev-CRM1 interaction network exports viral RNA into the cytoplasm with the help of RanGTP. Thus, viral RNA-bound Revs hijack host CRM1 to export viral RNA via NPCS [10,135,136,137]. Recent work has suggested that other NES-containing proteins, such as RNA binding protein 14 (RBM14) [138] and Phosphofurin acidic cluster-sorting protein 1 (PACS1) [139], could serve as Rev co-factors and play a crucial role in the nuclear export of viral RNAs mediated by Revs. In addition, several cellular factors have also been identified to regulate the nuclear export of HIV-1 mRNA, including the mRNA export adaptor protein Aly/REF and the RNA helicase DDX3 [140]. Further understanding of the molecular mechanisms underlying HIV-1 mRNA nuclear export could provide new insights into the development of therapies for HIV-1 infection.

Host CRM1 is also used to export viral RNA via the Rex protein of Delta retrovirus HTLV-1, which causes T-cell leukemia. Similar to the Rev protein, Rex binds to the RxRE (Rex response element) on unspliced and spliced viral RNA and interacts with CRM1 to export viral RNA into the cytoplasm with the help of RanGTP [140,141]. Similarly, the retrovirus MMTV (mouse mammary tumor virus) encodes the Rem protein, which interacts with viral RNA and CRM1 to export viral RNA to the cytoplasm [141,142,143]. Interestingly, the host adapter of CRM1, HuR, has also been used for the export of viral RNA by foamy viruses [144]. CRM1 is also involved in exporting IAV RNA by interacting with viral NP (with viral RNA complex) in the presence of RanGTP [120,145,146]. Thus, CRM1 is targeted by viral proteins for its advantages (i.e., the nuclear export of its RNA). Since CRM1 is associated with the nuclear export of mRNA, rRNA, and snRNA, viruses that use CRM1 to help export their own RNA to the cytoplasm can inhibit the export of host RNA and thus downregulate the expression of host genes. Viruses can deceive the host’s immune system by enhancing viral mRNA nuclear export to promote their cytokine gene expression. In addition, viruses can promote the export of viral mRNAs that encode inhibitors of host caspases to block cell apoptosis. Thus, viruses hijack the host’s mRNA nuclear export mechanism to help them trigger a successful infection and its subsequent spread [147].

### 4.2. The Virus Targets the Host mRNA Export Receptor NXF1

Similar to CRM1, the mRNA export receptor NXF1 is also a target for viral proteins exported by viral RNA (Figure 3B). NXF1 binds to CTEs (constitutive transporter elements). The NXF1-interacting protein NXT1 [148] further enhances this binding. Thus, unlike host mRNA, which requires the adapter protein to interact with NXF1, viral RNA interacts directly with NXF1 and is transported to the cytoplasm, thus having an advantage over the host system. The CTEs, which are located in the stem-loop RNA, are responsible for the nuclear export of the unspliced or incompletely spliced RNA of Mason-Pfizer monkey virus (MPMV) or type D retrovirus [14,149,150]. Similarly, other retroviruses, such as avian sarcoma and the leukemia virus, have used CTEs to export unspliced viral RNA [149,151]. Gamma retroviruses, including murine leukemia virus (MLV) and heterophilic murine leukemia virus-associated virus (XMRV), also use NXF1 for viral RNA export [152]. The nuclear export of MLV unspliced RNA is co-regulated by NXF1 and CRM1 pathway [153]. This export is mediated through NXF1’s interaction with the CAE (cytoplasmic accumulation element) of viral RNA [154]. Certain IAV mRNAs (e.g., viral mRNAs encoding M1 matrix proteins and M2 ion channels) also depend on NXF1 for export. The viral NS1 protein acts as an adapter (interacting with viral RNA and NXF1) for the nuclear export of viral RNA [16]. Using the auxin-induced deoxyribonucleic acid (AID) system, a recent study by Bhat et al. demonstrated that the nuclear export of influenza virus mRNAs is dependent on the TREX-2 complex. By labeling different components in mRNA and TREX-2 complex or targeting specific genetic components, they successfully revealed the interaction between the extranuclear transport process of influenza virus mRNA and the TREX-2 complex. Further RNA sequencing techniques have demonstrated the importance of this mode of transport in influenza virus extrinsic replication and infection [155]. The NP (which binds to viral RNA) of Ebola virus (EBOV, an RNA virus) also interacts with NXF1 to export viral RNA into the cytoplasm [118,156,157]. The infected cell protein 27 (ICP27) (which binds to viral RNA) of the herpes simplex virus 1 (HSV-1) interacts with NXF1 to achieve viral RNA export [118,158]. Similarly, the herpesvirus saimiri (HVS) TIP protein also interacts with NXF1 [156]. NXF1, therefore, is primarily a viral protein targeting viral RNA export. Since NXF1 is involved in the nuclear export of a large amount of mRNA in the host body, as described above, the viral hijacking of NXF1 can severely block the export of the host mRNA and thus block the expression of the host gene. AdV early protein E1B-55K, along with its cellular partner, E1B-55K associated protein 5 (E1B-AP5), interacts with viral mRNAs and facilitates the recruitment of NXF1–NXT1, which in turn stimulates the nuclear export of mRNP through the TREX-2–NPC interaction [157,159]. Hepatitis B virus (HBV), a member of the *Hepadnaviridae* family, can cause liver disease. The genetic material of HBV is composed of partially double-stranded DNA. The transcripts produced by HBV can be categorized into three distinct groups based on their splicing status and genomic location: unspliced pg (pregenomic) and preC (precore) mRNAs, spliced pg and preC mRNAs, and intronless subgenomic mRNAs [160]. The post-transcriptional regulatory element (PRE), a cis-acting RNA element that can be divided into two sub-elements, PRE 1 and 2 (SEP1 and SEP2), is present in all transcripts of HBV [161]. The zinc finger CCCH domain-containing protein 18 (ZC3H18) has been identified as binding SEP1 and recruiting components of the TREX complex for SEP1 nuclear export through the NXF1–NXT1 pathway [162]. TAR DNA binding protein (TARDBP) has been suggested to play a key role in the export of unspliced pgRNA isoforms [163]. The HBV X protein interacts with UAP56, which is potentially functional in the nuclear export of single-exon HBV mRNAs [164]. With a better understanding of the nuclear export mechanisms utilized by HBV mRNA, it may be possible to create novel targeted antiviral therapies that are required for the management of Hepatitis B.

To summarize, viruses have developed diverse strategies to manipulate the cellular pathways of mRNA nuclear export. The mechanisms utilized by viruses to facilitate their mRNA nuclear export are extremely variable, even within the same family (Table 2). Current and forthcoming research endeavors are expected to provide a comprehensive understanding of the molecular mechanisms of viruses’s mRNA nuclear export, which will provide novel anti-viral therapeutic strategies by targeting virus mRNA nuclear export.

## 5. Conclusions and Future Perspectives

To prevent the expression of mRNA encoding antiviral factors and to favor the competition with host mRNA in translation mechanisms, various research groups have identified ways by which distinct viruses impede the nuclear export of cellular mRNA. Overall, this leads to the inhibition of host protein translation. In addition, viruses have significantly affected our understanding of the nuclear export of cellular mRNA and the key roles involved.

The significance of these pathways in both proviral and antiviral mechanisms is underscored by these findings. These studies suggest that the virus invests many resources to disrupt nucleo-cytoplasmic transport, and that the host can antagonize these effects. Therefore, the nuclear import/export pathway is an attractive target for the development of novel antiviral therapeutics. An anti-IAV compound screening by targeting the nuclear export of IAV M mRNA has been reported by Esparza et al [172]. To effectively nuclear export of the viral mRNA, viruses utilize multiple and complex strategies to hijack cellular mRNA nuclear export machinary. To date, the mechanisms about these strategies still remain incomplete discovered, sparking considerable interest among researchers. Gaining a deeper understanding on the mRNA nuclear export of various types of viruses will offer new insights into both the virus replication process and our knowledge about cellular mRNA nuclear export mechanisms. 

## Figures and Tables

**Figure 1 ijms-24-12593-f001:**
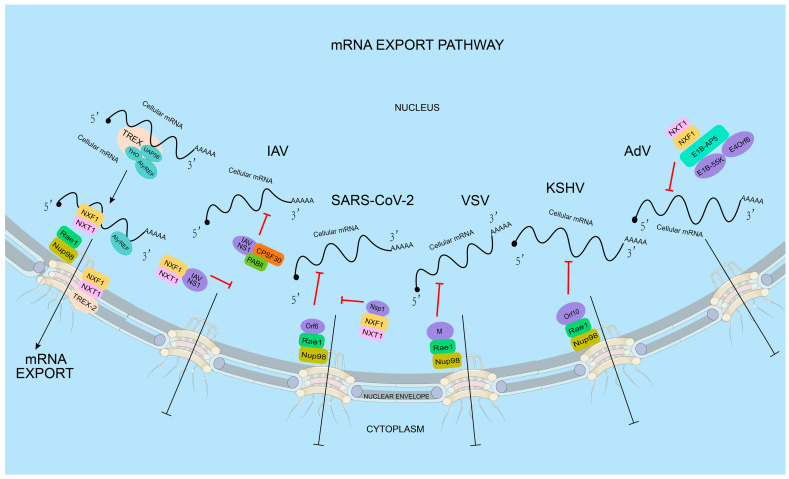
Viruses utilize multiple strategies to suppress host mRNA nuclear export. After transcription and splicing, the mRNA nuclear export factors start binding to the mRNA. Several factors are involved in the process, including the TREX complex, Aly/REF, and UAP56. The NXF1–NXT1 heterodimer is a major mRNA nuclear export factor that recruits mRNA to dock into the NPC. After several steps of translocation, the mRNA is exported from the nucleus to the cytoplasm for translation into protein. This process is targeted by viral proteins. IAV NS1 binds to CPSF30 and PAB II, resulting in the deficiency of mRNA polyadenylation. In addition, IAV NS1 can block the host mRNA nuclear export by directly binding to NXF1, inhibiting the host mRNA docking to the NPC. Several SARS-CoV-2 proteins could be involved in the inhibition of host mRNA nuclear export. SARS-CoV-2 Nsp1 directly binds to NXF1, preventing NXF1 from docking at the NPC. SARS-CoV-2 Orf6 forms a complex with the Rae1-Nup98 heterodimer, and its C-terminus occupies the mRNA binding groove in the Rae1-Nup98 complex. VSV and KSHV also block the host mRNA nuclear export by targeting the Rae1-Nup98 complex by VSV M, or KSHV ORF10 directly binds to the Rae1-Nup98 complex. AdV infection inhibits NXF1-mediated nuclear export of cellular mRNA. The AdV early protein E1B-55K, which cooperates with the AdV E4orf6 protein, recruits E1B-AP5 to bind NXF1.

**Figure 2 ijms-24-12593-f002:**
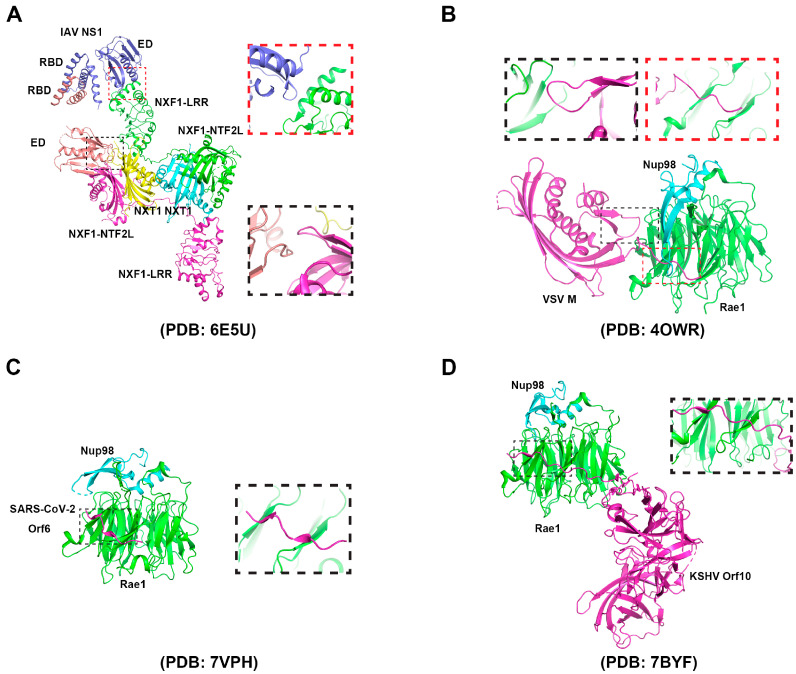
Structure of the viral protein and mRNA nuclear export factor complex. (**A**) Crystal structure of the IAV NS1 and NXF1–NXT1 complex. In the crystal, two NXF1–NXT1 heterodimers are linked by the two NXT1 molecules. Two NS1 molecules dimerize through their RBD domains. An effector domain binds to the LRR domain of NXF1. Another effector domain binds to the NTF2L domain of the neighboring NXF1, which is also the nucleoporin binding domain. (**B**) Crystal structure of VSV M and Rae1-Nup98 complex. The association of VSV M with Rae1-Nup98 is mainly due to the interaction of two extensions of the M global domain with Rae1. The highly conserved Methionine 51 facilitates interaction with Rae1. (**C**) Crystal structure of SARS-CoV-2 Orf6 and Rae1-Nup98 complex. This model shows that the SARS-CoV-2 Orf6 C-terminal domain and Rae1-Nup98^GLEB^ form a heterotrimer in which the Orf6 C-terminal domain binds directly to Rae1. The highly conserved methionine 58 of Orf6 plays an essential role in the interaction and is located at the mRNA binding site of Rae1. (**D**) Crystal structure of the KSHV ORF10 and Rae1-Nup98 complex. KSHV ORF10-Rae1-Nup98^GLEB^ forms a heterotrimer, and the KSHV ORF10 C-terminal domain interacts directly with Rae1. The conserved residues of ORF10 methionine 413 and leucine 60 majorly contribute to the interaction with Rae1.

**Figure 3 ijms-24-12593-f003:**
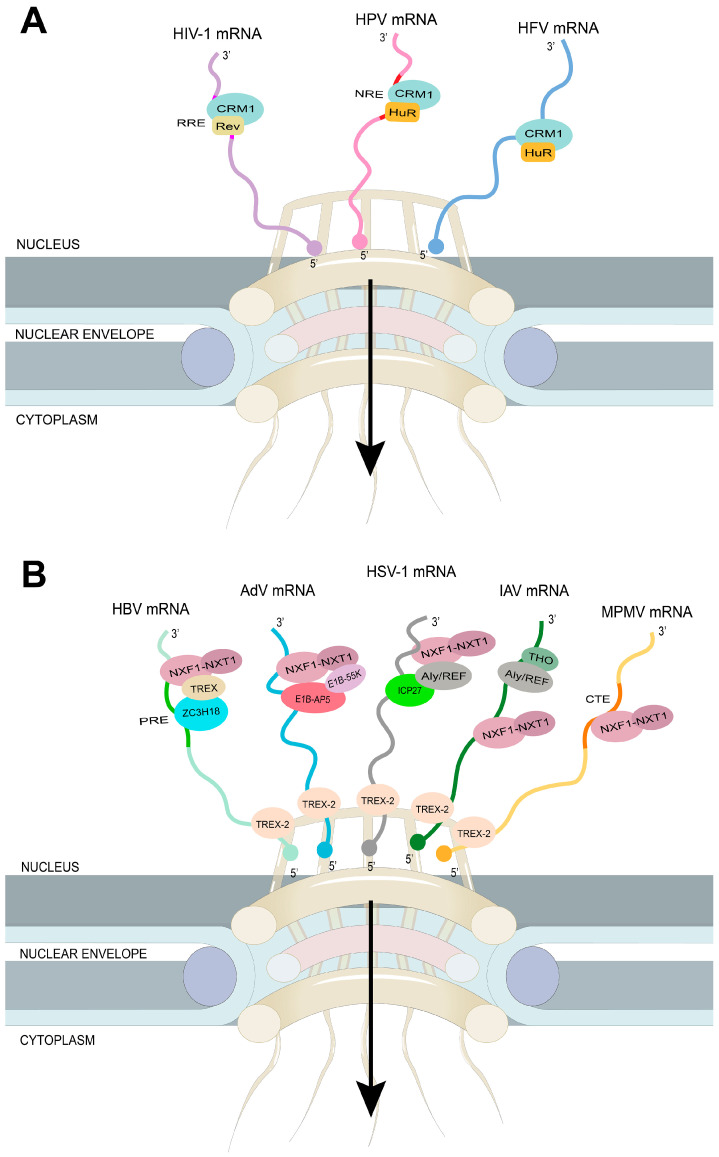
Viruses hijack the cellular machinery to facilitate the nuclear export of viral mRNA. (**A**) Examples of viral mRNA nuclear export dependent on the CRM1 pathway. Nuclear export of HFV mRNA, HPV mRNA, and unspliced human HIV-1 mRNA depends on the CRM1 pathway. The RRE RNA element is recognized by HIV Rev protein, which recruits the CRM1 for mRNA nuclear export. The nuclear export of HPV and HFV depends on the indirect recruitment of CRM1 by the cellular protein HuR. HuR can bind to the negative regulator element (NRE) RNA element on HPV mRNA. The mRNA-protein complex binds to RanGTP and is then translocated via NPC. (**B**) Examples of viral mRNA nuclear export dependent on NXF1–NXT1 mediating pathway. The nuclear export of HBV mRNA, AdV mRNA, HSV-1 mRNA, IAV mRNA, and MPMV mRNA is mediated by NXF1–NXT1 pathway. The PRE is recognized by cellular proteins, such as the ZC3H18, which is responsible for recruiting members of the TREX complex. The nuclear export of AdV mRNA involved the participation of E1B-55K and E1B-AP5. E1B-AP5 plays the role of interacting with both viral mRNA and NXF1. HSV-1 ICP27 is an export adaptor that links NXF1 and Aly/REF to the viral mRNA for nuclear export. The nuclear export of different types (intronless, unspliced, and spliced) of IAV mRNAs through the NXF1–NXT1 pathway is regulated by various cellular factors, including NS1-BP, UAP56, DDX19, and eIF4A3. The constitutive transport element (CTE) on MPMV mRNA strongly interacts with NXF1 for nuclear export.

**Table 1 ijms-24-12593-t001:** Mechanisms of viruses’ arrest of host mRNA nuclear export.

Group	Virus (Family)	Viral Protein	Mechanism
RNA-Virus	Influenza A virus (*Orthomyxoviridae*)	NS1	Disruption of NXF1 docking to the NPC [60]
		NS1	Inhibition of mRNA polyadenylation [60]
	SARS-CoV (*Coronaviridae*)	NSP1	Alteration of Nup93 localization [118]
	SARS-CoV-2 (*Coronaviridae*)	NSP1	Prevention of NXF1 docking at NPC [100]
		Orf6	Interaction with Rae1-Nup98 complex [119]
	Vesicular stomatitis virus (*Rhabdoviridae*)	M	Hijacking the Rae1-Nup98 complex [61,120]
	Poliovirus (*Picornaviridae*)	2A^pro^	Degradation of Nup62, Nup98, and Nup153 [61,120]
	Human rhinovirus (*Picornaviridae*)	3C^pro^	Degradation of Nup153, Nup214, and Nup358 [61,120]
	Encephalomyocarditis virus (*Picornaviridae*)	L	Altering the RanGDP/GTP gradient [76,77,78,79]
	Theiler’s Murine encephalomyelitis virus (*Picornaviridae*)	L	Hyper-phosphorylation of Nup98 [77]
	Rhesus rotavirus (*Reoviridae*)	NSP3	Hyperadenylation of cellular mRNA [121]
	Zika virus (*Flaviviridae*)		Degradation of UPF1 [80]
	Rift Valley fever virus (*Phenuiviridae*)	NSs	Unknown [122]
DNA-Virus	Adenoviruses (*Adenoviridae*)	E1B-55K, E4Orf6	Interaction with E1B-AP5 [112]
	Kaposi’s sarcoma-associated herpesvirus (*Herpesviridae*)	ORF10	Hijacking the Rae1-Nup98 complex [116]

**Table 2 ijms-24-12593-t002:** Mechanisms of viruses’s mRNA nuclear export.

Group	Virus (Family)	Pathway	Mechanism
RNA-Virus	Influenza A virus (*Orthomyxoviridae*)	NXF1–NXT1	Regulation by viral protein NS1, and cellular proteins NS1-BP, UAP56, DDX19, and eIF4A3 [14,28,29,30].
	Human Immunodeficiency Virus 1 (*Retroviridae*)	CRM1	The recruitment of CRM1 by HIV-1 Rev [10,135,136,137].
	Mason-pfizer Monkey virus (*Retroviridae*)	NXF1–NXT1	The recruitment of cellular export receptor NXF1–NXT1 by constitutive transport elements [148,149,150].
	Human T-cell leukemia virus type 1 (*Retroviridae*)	CRM1	The recruitment of CRM1 by HTLV-1 Rex [165,166].
	Murine leukemia virus (*Retroviridae*)	CRM1	Unknown (unspliced mRNAs for packaging) [152].
		NXF1–NXT1	Dependent on UAP56 (spliced mRNA) [152].
			Dependent on THOC5, THOC7, and SRp20 (unspliced genomic mRNA) [152,167].
	Rous Sarcoma Virus (*Retroviridae*)	CRM1	The contribution of RSV Gag (unspliced mRNAs) [168].
		NXF1–NXT1	The recruitment of NXF1 and DDX5 by direct repeat sequences flanking the *src* gene [154].
	Prototype foamy virus (*Retroviridae*)	CRM1	The recruitment [163] of CRM1 by the host adapter HuR [169].
DNA-Virus	Adenoviruses (*Adenoviridae*)	NXF1–NXT1	Regulation by AdV E1B-55K, and cellular protein E1B-AP5 [112].
	Herpes simplex virus 1 (*Herpesviridae*)	NXF1–NXT1	The interaction of HSV-1 ICP27 with NXF1, Aly/REF, and viral mRNAs [118,158].
	Human papillomavirus (*Papoviviridae*)	CRM1	The recruitment of host adapter HuR by viral AU rich element [170].
	Hepatitis B virus (*Hepadnaviridae*)	NXF1–NXT1	The recruitment of TREX components by ZC3H18 (intronless mRNA) [162].
			The recruitment of NXF1 by an unknown cellular factor (unspliced PreC mRNA) [171].

## Data Availability

Not applicable.

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
