# Peer review of "Virus Infection and mRNA Nuclear Export"

_ijms, 2023, doi:10.3390/ijms241612593_

Round 1

Reviewer 1 Report

This manuscript was to review the literature on nuclear export modulation by viruses. The particular focus was not stated beyond a general overview. As a result, the article is diffuse and superficial and adds little to the literature already published.

The authors do bring up newer citations, but do not reveal how the new experiments were carried out. They make major conclusions, like the virus is diminishing host antiviral function without saying the specifics. As a result, I learned the authors' opinion of the work, but not the data that cemented their conclusion of specificity. 

As a result, these is little new information in this article to warrant its addition to the literature, unfortunately.

Some typos and misstatements undermine the overall superficial coverage to the literature. 

Author Response

We greatly appreciate the reviewer's constructive suggestions. In response to the suggestions, we have made the following additions: The experiments and methods mentioned in the recent work have been included in the sections titled 'Viral Infection Inhibits the Nuclear Export of Host mRNA' (Pages 15-16, lines 323-334 for influenza virus NS1 protein; Page 17, lines 354-356 for VSV M protein; Page 18, lines 380-386 for SARS-CoV-2 Nsp1 protein; Page 21, lines 443-451 for KSHV Orf10 protein), and 'The Nuclear Export of Viral mRNA' (Page 27, lines 562-570 for influenza virus mRNAs)

We have worked on both the language and the readability and also involved native English speakers for language corrections.

Reviewer 2 Report

ijms-2385544

Virus Infection and mRNA Nuclear Export

Jiayin Guo, Yaru Zhu, Xiaoya Ma, Guijun Shang, Bo Liu and Ke Zhang

This review describes the mechanisms by which viruses disrupt host mRNA nuclear export during infection and the key strategies that viruses use to promote their mRNA nuclear export. The authors think these knowledges reveal novel antiviral strategies to inhibit viral mRNA transport and increase host mRNA export.

This review well summarizes virus mRNA export in relation to host RNA export pathways. I have only one suggestion to improve this manuscript.

For the host RNA export, the length of RNA plays important roles in determining which pathway the RNA takes. It would be nicer if the authors could introduce Dr. Ohno’s group works more in host RNA export section. For example, their works for hnRNP C in determining RNA length should be included (McCloskey A, Taniguchi I, Shinmyozu K, Ohno M. Science. 2012 Mar 30;335(6076):1643-6. doi: 10.1126/science.1218469., Dantsuji S, Ohno M, Taniguchi I. Nucleic Acids Res. 2023 Feb 22;51(3):1393-1408. doi: 10.1093/nar/gkac1250.).

English writing is fine.

Author Response

We greatly appreciate the reviewer's suggestion to consider the choice of the nuclear export pathway, which also depends on the length of the RNA. Following the reviewer's advice, we have described the relationship between RNA length and the selection of the export pathway. Additionally, we have cited the relevant paper in the section on Cellular mRNA nuclear export (page 10, lines 198-210).

Round 2

Reviewer 1 Report

line 96-97 "of viral mRNA meanwhile inhibiting the transport of host mRNA to prevent an appropriate host immune response" ref [12-14] belong to the first part of this sentence, and not here.

this conclusion is complete conjecture, there is not a citation to document this information and this needs to be fixed. 

Lines 98-120 are background info and there is nothing added to explain virus interplay with any of the steps, thus nothing new is provided.

"Although many host cell 128 processes are involved in viral replication and assembly, we discuss below the 129 mechanisms by which viruses hijack or usurp host messenger RNA export 130 through RNA exporting viral gene expression, replication, assembly, and 131 inhibiting host messenger RNA export, thereby reducing host gene expression 132 and causing a successful and sustained infection by the immune response." IS THIS ANSWERED?

Please correct, DNA is transcribed, not RNA 137 eukaryotic cells, mRNA transcription occurs in the nucleus, and the newly 138 synthesized mRNA molecules must be exported to the cytoplasm for translation 139 to occur

lines 235-238 have incomplete sentences. Please eliminate the parenthesis, if if its worth saying it does not require ().

Line 199-forward is much better, but some phrases are unclear and need to be explained in more detail. See Underlined phrases and improve detail:

McCloskey et al. showed that 200 heterologous nuclear ribonucleoprotein (hnRNP) C tetramer has the function of 201 measuring RNA length for the classification export of RNA polymerase II 202 transcripts [54]. It has been demonstrated that hnRNP C tetramers play a key 203 role in the identification and measurement of RNA length, which can be used 204 to distinguish between different processes and RNA destinations in RNA 205 biosynthesis. Dantsuji et al. have proven that the underlying mechanism is that 206 the hnRNP-C tetramer binds the cap-binding complex (CBC) on mRNA and 207 blocks PHAX recruitment, so as to classify (what is meant by classify) the transcripts of RNA polymerase 208 II [55]. This discovery OF WHAT....NEED TO DEFINE CONTEXT contributes to a deeper understanding of RNA formation 209 and function and provides new ideas and potential strategies for future disease 210 treatments.

Recent studies have shown that mRNA nuclear export is also a 211 tightly regulated process that is subject to various cellular and environmental 212 cues. For instance, stress signals, such as viral infections or heat shock, can 213 alter the efficiency of mRNA nuclear export. SO IT FOLLOWS VIRUSES ARE NOT CHANGING EXPORT, ITS THE SHOCK AND STRESS RESPONSE, PLEASE EXPLAIN

Additionally, mutations in some of 214 the export factors have been linked to various human diseases, such as cancer 215 and neurodegenerative disorders [56, 57].  AND WHAT DOES THIS TELL US ABOUT VIRUS?

TYPO: 256 causes protein degradation in nucleoporins including NUP62, NUP98, and 257 NUP153, leading to changes in the NPC structure and facilitating host mRNA 258 export [66-72]. FACILITATING OR WHAT?

TYPO PICORNAVIRUSES:  260 NPC and mRNA export [73]. Unlike these viruses, picorbonucleoviruses such 261 as Tyler Murine encephalomyelitis virus (TMEV) and encephalomyocarditis 262 virus (EMCV) lack proteolytic activity in 2APro

LINES 278 and produces long non-coding RNA (lncRNA) called SfRNAs, which are 300- 279 400 nucleotides long, to stabilize short-lived host mRNA and facilitate its 280 replication [78]. WHAT DOES THIS ENTAIL? WHAT HOST RNA ARE THE TARGETS?

315-400 nicely improved

401 disease [103]. The genome of human AdVs is quite dense, with more than 40 genes COMPLETE THE SENTENCE... IN THE SPAN OF X BASES [104].

491 block host cell apoptosis strategies to get rid of virus-infected cells Error! 492 Reference source not found..

HIV IS DEFINED TWICE INSTEAD OF ONCE LINE 509

Thus, the HIV-1’s mRNA 515 should be exported to the cytoplasm for translation. The Rev protein contains 516 the leucine-rich NES domain, which is recognized by CRM1. In addition, the 517 Rev protein interacts with RRE (Rev Response Element, a highly structured 518 RNA), which is often unspliced PERIOD.

REV (NOT RRE) and also acts as an arginine-rich NLS (Nuclear 519 Location Signal) [10, 134-136].

WORK S IS A TYPO LINE 522

LINE 540 foamY

534 RNA and interacts with CRM1 to export viral RNA into the cytoplasm with the 535 help of RanGTP Error! Reference source not found.Error! Reference 536 source not found..

Similarly, a double retrovirus MMTV (Mouse 537 MammaryTumor Virus) e

Since CRM1 is 544 associated with the nuclear export of mRNA, rRNA, and snRNA, viruses that 545 use CRM1 to help export their own RNA to the cytoplasm down-regulate the 546 export of host RNA (and thus down-regulate the expression of host genes).  WHAT IS THE CITATION FOR THIS? THIS IS COMPLETE CONJECTURE WITHOUT DATA AND SHOULD BE REMOVED OR EXPLAINED IN DETAIL

TYPOS LINE 556 557

TYPOS 549 proteins exported by viral RNA. NXF1 binds to Ctes (Constitutive Transporter 550 Simple (Figure 3B)). A retrovirus, the stem ring RNA of unspliced or 551 incompletely spliced RNA of MPMV (Mason-Pfizer Monkey Virus) or type D 552 retrovirus, is exported to the cytoplasm 

Gamma retroviruses, including MLV (Murine Leukemia Virus) 559 and XMRV (heterophilic murine leukemia virus-associated virus), also use 560 NXF1 for viral RNA export. NO CITATIONS

NEED TO INCLUDE THE NEWER WORK OF:

NXF1 and CRM1 nuclear export pathways orchestrate nuclear export, translation and packaging of murine leukaemia retrovirus unspliced RNA.

Mougel M, Akkawi C, Chamontin C, Feuillard J, Pessel-Vivares L, Socol M, Laine S.RNA Biol. 2020 Apr;17(4):528-538. doi: 10.1080/15476286.2020.1713539. Epub 2020 Jan 23.

REFERENCES ARE WRONG:

This export is mediated through NXF1's interaction 561 with the CAE (cytoplasmic accumulation element) of viral RNA [150]. Certain 562 IAV mRNAs (e.g., viral mRNAs encoding M1 matrix proteins and M2 ion 563 channels) also depend on NXF1 for export. The viral NS1 (Non-Structural 564 Protein 1) protein acts as an adapter (interacting with viral RNA and NXF1) 565 for the nuclear export of viral RNA [151]. 

A few items, ie workS, some incomplete sentences.

replace the use of words like 'splitting' and nuclear "blobs"  
